# Metacognition as a Consequence of Competing Evolutionary Time Scales

**DOI:** 10.3390/e24050601

**Published:** 2022-04-26

**Authors:** Franz Kuchling, Chris Fields, Michael Levin

**Affiliations:** 1Department of Biology, Allen Discovery Center at Tufts University, Medford, MA 02155, USA; franz.kuchling@tufts.edu; 223 Rue des Lavandières, 11160 Caunes Minervois, France; fieldsres@gmail.com; 3Wyss Institute for Biologically Inspired Engineering, Harvard University, Boston, MA 02138, USA

**Keywords:** metacognition, metaprocessor, coevolution, coadaptation, temporal scales, active inference, predator–prey models, coupled genetic algorithms, generative adversarial networks

## Abstract

Evolution is full of coevolving systems characterized by complex spatio-temporal interactions that lead to intertwined processes of adaptation. Yet, how adaptation across multiple levels of temporal scales and biological complexity is achieved remains unclear. Here, we formalize how evolutionary multi-scale processing underlying adaptation constitutes a form of metacognition flowing from definitions of metaprocessing in machine learning. We show (1) how the evolution of metacognitive systems can be expected when fitness landscapes vary on multiple time scales, and (2) how multiple time scales emerge during coevolutionary processes of sufficiently complex interactions. After defining a metaprocessor as a regulator with local memory, we prove that metacognition is more energetically efficient than purely object-level cognition when selection operates at multiple timescales in evolution. Furthermore, we show that existing modeling approaches to coadaptation and coevolution—here active inference networks, predator–prey interactions, coupled genetic algorithms, and generative adversarial networks—lead to multiple emergent timescales underlying forms of metacognition. Lastly, we show how coarse-grained structures emerge naturally in any resource-limited system, providing sufficient evidence for metacognitive systems to be a prevalent and vital component of (co-)evolution. Therefore, multi-scale processing is a necessary requirement for many evolutionary scenarios, leading to de facto metacognitive evolutionary outcomes.

## 1. Introduction

The idea of metacognition—“thinking about thinking”—is as old as introspective philosophy and is central to both cognitive neuroscience [1,2,3,4,5,6] and artificial intelligence [7,8,9,10,11]. Metacognition is sometimes defined to include exclusively human capabilities, e.g., theoretical self-knowledge [12]. In its simplest form, however, any regulator that is an integral part of some larger system can be viewed as a “metacognitive” model of the lower-level system components that it regulates [13]; systems that include such internal regulators display self-monitoring and self-regulation, the most basic attributes of metacognition noted in [12]. In humans, metacognition is broadly associated with executive control [14,15] and “deliberate” Process-1 problem solving [16,17,18], though whether there is a clean, architectural difference between this and “automated” Process-2 problem solving remains subject to considerable debate [19,20,21]. It is clear, in particular, that metacognitive processes are not always consciously introspectable. How such “high-level” metacognition relates to more prosaic forms of regulation, either in humans or across phylogeny, remains poorly understood and indeed relatively uninvestigated.

Here, we adopt a broad notion of metacognition, employing this term to indicate the function of regulating some “lower-level” computational process. An architectural component that implements metacognition will be referred to as a metaprocessor for the component being regulated. This purely functional definition of metacognition explicitly avoids the seemingly intractable philosophical problem of determining whether a given system is “cognitive”; our approach to this latter problem is detailed elsewhere [22]. It also explicitly depends on how the larger system of which the metaprocessor is a component is identified. In particular, this larger system must be identified as comprising both the metaprocessor and the system that it regulates. A thermostat, for example, can be considered a metaprocessor for the on/off switch of a furnace, provided that the thermostat, the switch, and the rest of furnace are considered to compose a single system, a “regulated furnace”; if the thermostat is considered independently of this larger system, it is simply a thermally regulated switch. The question of whether something is a metaprocessor is, therefore, not an ontological question, but rather an architectural question that depends on how the larger system that contains it is functionally decomposed.

With this understanding of metacognition as a function and metaprocessors as components, we address the question of why natural selection has produced systems capable of metacognition, i.e., systems regulated by metaprocessors. We suggest that metacognition can be expected whenever (evolutionary) fitness functions—as we will discuss in the context of active inference in Section 2.1 and in genetic algorithms in Section 3.2.3—vary on multiple timescales. As we will see, such fitness functions are crucially multi-scaled. Metacognition, in this view, provides a context-dependent switch between depth-first and breadth-first searches that allows the avoidance of local minima. As such, it can be expected to be both ancient and ubiquitous. We can, indeed, identify slowly acting regulators of faster control systems even in prokaryotes; e.g., acetylation that changes the sensitivity of the chemotaxis “switch” molecule CheY in *E. coli* [23]. This example reminds us that biological systems implement memories at multiple scales using various molecular, cellular, and bioelectric substrates [24].

In what follows, we first briefly review comparative studies that take human metacognition as a starting point in Section 2.1. We outline the computational resources required to implement metacognition in arbitrary systems, focusing on requirements for memory and an ability to measure duration, in Section 2.2. We review the conditions under which interactions between physical systems are mediated by Markov blankets (MBs) [25,26] in Section 2.3, and then provide a general overview of the framework of active inference to minimize variational free energy (VFE) as it applies to living systems from the cellular scale upwards [27,28,29,30,31] in Section 2.4. As this framework is provably general for systems that maintain states conditionally independent from those of their environments (or in quantum-theoretic terms: separable states) over macroscopic times [32,33], it provides a general formalism for discussing the evolution of computational architectures enabling metaprocessing. We turn in Section 3 to our main hypothesis: that metaprocessing, and hence metacognition as a function, can be expected to arise in any systems faced with selective pressures—effectively, sources of VFE—that vary on multiple timescales. We begin in Section 3.1 by proving, under generic assumptions, that metacognition is more energetically efficient, and hence confers higher fitness, than purely object-level cognition when selection operates at multiple timescales. We then turn in Section 3.2 to illustrations of this result, reviewing simulation studies of a variety of architectures, including multi-agent active inference networks, Lotka–Volterra-based systems of predator–prey interactions, coupled genetic algorithms, and coupled generative adversarial networks (GANs), showing how multiple timescales intrinsic to the relevant problem domains induce metacognition as a high-level function. We then discuss in Section 3.3 the emergence of coarse-graining representations of both time and space as a general consequence of metacognition. We conclude that metacognition is far from being human- or even mammal-specific and is not a special case; rather it can be expected at every level of biological organization and always addresses fundamentally the same problem, that of dealing effectively with uncertainties having different spatio-temporal scales.

## 2. Background

Before we investigate metacognition as an intrinsic function in generic terms, as well as in a variety of well-studied model paradigms, we first consider metacognition from an evolutionary perspective, outline the key computational resources necessary for metaprocessing, and briefly review the active inference framework as a provably general model of systems capable of metacognition.

### 2.1. Metacognition from an Evolutionary Perspective

The idea of metacognition as a function has two distinct theoretical antecedents: the philosophy and, later, psychology of deliberate, conscious control in humans, and the practical, engineering development of regulators and control systems. The definition of “conscious control” is of course problematic, as there are well-known and substantive disagreements about what consciousness is and how it is manifested (see for example [34,35] for recent reviews). Global workspace [36,37] and higher-order thought [38,39] approaches to consciousness naturally involve metacognition; others, for example Integrated Information Theory [40], do not. Even the notion of content of consciousness is controversial [41]. It is, moreover, as noted earlier, not clear what distinguishes “deliberate” from “automatic” processes [19,20,21]. These definitional controversies motivate our current purely functional approach to metacognition. This approach is broadly consonant with the systems engineering tradition, particularly as elaborated within artificial intelligence and robotics, where canonical metacognitive functions such as intrinsic motivation [42], curiosity-driven allocation of learning resources [43], and ethics-driven decision making [44] have become increasingly prominent components of autonomous or semi-autonomous systems.

Broadening the definition of metacognition away from human-specific capabilities and avoiding debates about consciousness allows explicit consideration of metacognition as an evolutionary development with potentially deep phylogenetic roots (but also see [22] for an approach to consciousness fully consistent with this evolutionary perspective). As expected, given the self-monitoring and self-regulating components of metacognition, canonical metacognitive tasks such as introspection, deliberate choice, and voluntary recall engage components of both executive [15] and default-mode [45] networks [46,47,48,49,50]. Such tasks can be translated to nonhumans by removing requirements for verbal communication and employing species-appropriate measures of choice, risk, confidence, and self-representation [51]. The involvement of homologous cortical structures in metacognition tasks in humans and other mammals provides additional support for human-like metacognition in other primates [52,53,54,55,56] and rodents [57]; claims of metacognitive ability based on behavioral tests and observation in the wild remain controversial [51,58,59]. Whether metacognitive abilities have been developed, presumably by convergent evolution, in invertebrates, e.g., cephalopods [60,61], also remains controversial.

Here, we offer an alternative approach to metacognition that focuses not on behavioral tests of function, but rather on the presence of architectural structures supporting metaprocessing. This architecture-driven approach avoids a priori assumptions derived from human uses of metacognition; hence, it leaves open the possibility of metacognitive functions in phylogenetically more diverse organisms that might not be noticed or recognized using human cognition as an implicit definition. This leads us to ask, from an abstract, architectural perspective, what computational resources are required to support metaprocessing.

### 2.2. Computational Resources for Metaprocessing

A generic metaprocessing architecture is shown in Figure 1: the metaprocessor samples both the input to and internal representations within an “object-level” processor and provides outputs that either regulate the object-level process or modify its output. Note that in this representation, meta- and object-level processors are regarded as components of a single containing system that receives inputs and produces outputs. Communication between object- and meta-level processors occurs via a defined channel that can, from a design perspective, be considered an application programming interface (API) and is restricted to the data structures supported by this channel. The Good Regulator Theorem requires that the metaprocessor be (i.e., encode or implement) a “model” of the object processor [13]. The regulatory capability of a metaprocessor is subject to the usual tradeoff between the accuracy, complexity, and computational efficiency of its model; overparameterized models or models that require too much time to execute are clearly nonoptimal. Barring these, regulatory capability increases as access to the internal state of the object processor increases, with the coding capacity of the communication channel as a hard limit. Increased access clearly requires increased computational and memory resources for the metaprocessor. From a design perspective, there is a tradeoff between the algorithmic and computational resources allocated to object- and meta-level processing. Even in a technological setting, the object level can become “frozen” for reasons outside a designer’s control, forcing the metaprocessor to take on tasks beyond simply monitoring and regulating the performance of the object processor. As evolution tends to proceed by “frozen accidents,” we can expect metaprocessors taking over what would, from a design perspective, be regarded as object-level tasks, to be ubiquitous in the architectures of living systems. Epigenetic regulation can, for example, be viewed as a meta-level regulator of gene expression that “takes over” the role ordinarily played by transcription factors when the latter cannot evolve either fast enough or with sufficient specificity to meet a selective challenge. In what follows, however, we will focus on the more traditional role of metaprocessors as regulators, not partial replacers, of object-level functions.

The architecture shown in Figure 1 is clearly hierarchical. The metaprocessor “knows” about the object-level processor only the information encoded in its inputs; the object-level processor, similarly, “knows” about the metaprocessor only the information encoded in the metaprocessor’s outputs. Any such architecture is amenable to virtualization: either processor can be replaced with any other processor having the same I/O behavior. The architecture in Figure 1 can, therefore, be considered a simple (two-level) virtual machine hierarchy [62]; it can be extended into a hierarchy of arbitrary depth. A thermostat, for example, is insensitive to whether the gas furnace it has controlled is swapped out for an electric furnace, or even for a heat pump. Genetic engineering is, similarly, effective to the extent that the expression and function of a gene are only minimally and controllably dependent on the cellular environment into which it is placed.

The local memory available to the metaprocessor determines both the maximum resolution of its model of the object-level process and the maximum window of history data that it can maintain. Hence, local memory is a critical resource for performance-based regulation. In simple systems such as thermostats or the CheY system in *E. coli*, limited memory can be compensated for by a long characteristic time for regulatory changes; thermostats take advantage of the thermal mass of rooms, while CheY acetylation is much slower than phosphorylation. More sophisticated systems that regulate toward long-term goals, however, require sufficient memory, implementing appropriate data structures, to maintain records of performance history. Metaprocessors that regulate learning systems to maximize learning efficiency provide an example. Any learning algorithm can be considered a function:(1)$$L:\left(f,T\right)\to f^{\prime }$$
where f  is a function and T is a set of training data. Hence, any learning algorithm can be considered a metaprocessor. Autonomous systems in complex environments must, however, select their own training data by implementing an exploration strategy that focuses on nontrivial but learnable features of the task environment [63]. Such selection systems may be fixed—effectively hardwired—or support learning. Improving training set selection through learning requires a higher-level, longer-time-window metaprocessor that regulates training-set selection by measuring learning progress. Humans accomplish this task heuristically, as expected for systems with limited computational and memory resources [64].

Metaprocessors serving long-term goals typically employ input from multiple sources and coordinate multiple types of actions; autonomous vehicles provide an example [65]. In such systems, the metaprocessor effectively serves multiple object-level processors, regulating their joint behavior toward both near-term (e.g., collision avoidance) and longer-term (e.g., timely arrival at a destination) objectives. As resources become constrained, shared memory and input queuing can be expected to replace true parallelism on both input and output sides. Global workspace (GW) models of human attention allocation and executive control invoke such resource-limited solutions for integrative metaprocessing [36,37]; the LIDA architecture replicates a GW model for robotic control [66].

From a design perspective, metaprocessor architectures are typically explicitly hierarchical and metaprocessors are typically explicitly centralized. Nonhierarchical distributed-system architectures, for example ACT-R [67], lack explicit metaprocessors. Hierarchical recurrent architectures, for example ART [68], similarly exhibit no explicit metaprocessing. We will show below, however, that metaprocessing emerges generically as an effective or apparent function when the interaction between a system and its environment is characterized by fitness functions with multiple characteristic timescales. To show this, we require a suitably generic way to describe both interaction and fitness, to which we now turn.

### 2.3. Interaction across a Markov Blanket

Biological systems are finite, and their interactions with their environments exchange only finite quantities of energy. The most general representation of such finite interactions is bipartite: some finite system S interacts with a finite environment E that is defined to be everything other than S. This definition renders the joint system SE both finite and closed. We can, therefore, represent the interaction as in Figure 2a: the systems S and E interact via a Hamiltonian (total energy) operator $H_{SE}\;$ that is defined at the boundary **ℬ** separating S from E. Formally, the boundary **ℬ** is a decompositional boundary in the state space of the joint system SE; it separates states of S from states of E. The Hamiltonian $H_{SE}\;$ is, formally, a linear operator on this joint state space. The conservation of energy requires that the net energy flow between S and E is asymptotically zero, i.e., that the interaction is asymptotically adiabatic; we will assume for simplicity that it is adiabatic over whatever time scale is of interest.

We now make an explicit assumption that the states of the systems S and E are independently well-defined. This is always the case in classical physics; in quantum theory, it is the assumption that S and E are not entangled (i.e., the joint quantum state $|SE\rangle$ is *separable* and factors as $|SE\rangle =|S\rangle |E\rangle$) over the time period of interest. In this case, the interaction $H_{SE}$ can be represented, without loss of generality, as a sequence of exchanges of finite information between S and E, i.e., S writes a finite bit string on **ℬ**, which E then reads, and the cycle reverses [22,33,69]—see [70,71] for technical details. The information exchanged between S and E—hence the information encoded on **ℬ** in each interaction cycle—can be specified exactly: at each instant, **ℬ** encodes the current eigenvalue of the operator $H_{SE}$.

Under these conditions—finite interaction in the absence of quantum entanglement—the decompositional boundary **ℬ** functions as an MB between S and E [72]. An MB surrounding a system S is, by definition, a set of states $m=\left(s,a\right)$ such that internal states i of the blanketed system S (if any) depend on external or environmental states *e* only via their effects on m [25,26]. In practice, an MB exists whenever the state space of $S\left(E\right)$ includes states not on the boundary **ℬ**. Friston [27,28] introduced the decomposition of the MB into sensory states s and active states a, with the further constraint that sensory states are not influenced by internal states and active states are not influenced by external states; this distinction is illustrated in Figure 2b and, in two biological contexts, in Figure 3. Any MB is, clearly, completely symmetrical; the “sensory” states of E are the “active” states of S and vice versa. An MB generalizes the function of an API, in effect defining, and therefore limiting, the communication channel between any system S and its environment E.

Note that nothing in the above places any constraints on the internal interactions $H_{S}\;$ and $H_{E}$ of S and E, respectively. In particular, neither $H_{S}$ nor $H_{E}$ can be inferred from the interaction $H_{SE}$ defined at the boundary **ℬ** that implements the MB separating S from E. The existence of the MB thus places significant restrictions on what S can “know” about E and vice-versa. Because S has no direct access to the state e of E, S cannot measure the dimension of e, and hence cannot determine the number of degrees of freedom of E. The MB similarly prevents any direct access to the internal interaction $H_{E}$ of E. We can, therefore, construct any finite decomposition E=FG with internal interaction $H_{E}=H_{F}+H_{G}+H_{FG}$ without affecting what S can detect, i.e., without affecting the information encoded on the MB by $H_{SE}$ in any way. Hence, S cannot “know” about decompositions of E. Any “subsystems” of E represented by S are, effectively, sectors of the MB that are defined by computational processes implemented by S; see [22,33,69,73] for formal details and further discussion. As the MB is completely symmetrical, these considerations apply equally to E.

As emphasized by Friston [32], any system that occupies a nonequilibrium steady state (NESS), or more generally, has a well-defined NESS density, must be separated from its environment by an MB. This condition is, however, not necessary; indeed, no system with an MB could perform inferences or enact nontrivial behaviors if it were. Even with a fixed $H_{SE}$, changes in the eigenvalue encoded on the MB, and hence in the energy transferred across it, will generally correlate with changes in the internal states of both S and E. Time dependence in $H_{SE}$ will also induce time-varying states in both S and E (see [33] for further discussion). The Free-Energy Principle (FEP) [27,28,29,32] is the statement that closed joint systems SE will asymptotically approach a constant $H_{SE}$ and, hence, that both S and E have NESS densities as attractors; it is shown in [33] that this principle is, for quantum systems, asymptotically equivalent to the Principle of Unitarity, i.e., of the conservation of information within the joint system.

### 2.4. Active Inference Framework

The Bayesian inference framework enables quantitative models linking sensory mechanisms (i.e., inputs) with functional behaviors (i.e., outputs) in arbitrary classical [32] and quantum [33] systems. This framework has been applied extensively to biological systems [28,29,30,31,74,75,76,77,78]; here, we consider it more generally. The variational free energy (VFE) that is being minimized in Bayesian inference follows out of classical analytical and statistical physics considerations as a unique form of a least action principle. With the partition defined by the MB in hand, we can interpret internal states as parametrizing some arbitrary probability density $q\left(e\right)$ (its “beliefs”) over external states. This replaces the Lagrangian being used to compute the gradient descent in classical least action (or variational) principles with a VFE functional F of form (cf. Equation (8.4) in [32]):(2)$$\begin{array}{cccc} F\left(m,i\right)= & -\ln\;p\left(m,i\right) & + & D_{KL}(q\left(e\right)||p(e|m,i),\\ & \mathrm{Surprisal} & & \mathrm{Bound} \end{array}$$
where
(3)$$D_{KL}\left(p|q\right)=\overset{\infty }{\underset{-\infty }{\int }}p\left(x\right)\;\ln\frac{p\left(x\right)}{q\left(x\right)}\;dx$$
is the Kullback–Leibler Divergence defined as the expectation of the logarithmic difference between the probabilities p and q. This makes Equation (2) a logarithmic difference between the (variational) density or Bayesian “beliefs” about external states $q\left(e\right)$ and actual probability densities p(e|m,i) of external states given the MB state m and the internal state i. Note that the VFE is a measure of uncertainty about how the current behavior of the environment E (i.e., the external state e) will affect the future state m of the MB; the VFE is thus completely distinct from the thermodynamic free energy required to drive computation, as discussed in Section 3.1 below.

The first term in Equation (2) is known as (Bayesian negative log) model evidence, or marginal likelihood, describing the likelihood that the sensory inputs (i.e., the MB state m) were generated by the system’s generative model, implicit in the internal state i. Because negative evidence is equivalent to unexpected sensory inputs, this term is also referred to as surprisal. The second term is referred to as relative entropy, which minimizes the divergence between the variational and posterior densities $q\left(e\right)\;$ and $p\left(e|m,i\right),\;$ respectively. As a result, maximizing model evidence—by either changing i to encode new beliefs or acting on E so as to alter m—is equivalent to minimizing the VFE $F\left(m,i\right)\;$ of the system [28,29,30,32]. Because the divergence of the second term can never be less than zero, free energy is an upper bound on the negative log evidence. For this reason, the Kullback–Leibler term is also referred to as a bound on surprisal.

With this definition of VFE, the FEP can be stated as the principle that any system S with an MB will behave so as to minimize VFE, i.e., so as to maximize its ability to predict the future actions of its environment E on its MB. This is active inference [27,28,29,32]. Maximal predictive ability clearly implies minimal time variation in both $H_{SE}$ and $H_{S}$, i.e., S occupying or being very close to its asymptotic NESS. The converse is also true:

**Theorem** **1.**
*Time variation in*

$H_{SE}$

*increases VFE for*

$S\left(E\right)$



**Proof.** Time variation in $H_{SE}$ is time variation in the spectrum of $H_{SE}$, i.e., in the eigenvalues of $H_{SE}$ encoded on the MB. VFE measures the unpredictability of these eigenvalues; hence, variation in the eigenvalues increases VFE. □

Theorem 1 imposes an additional limit on what S can “know” both about E and about itself. Consider a time evolution $U=SE\to U=S^{\prime }E^{\prime }$ that “moves” the decompositional boundary **ℬ** within a fixed joint system U. This evolution corresponds to an exchange of degrees of freedom between S and E; S engulfing part of E or vice versa would be an example. Any such evolution changes the Hamiltonian, $H_{SE}\to H_{S^{\prime }E^{\prime }}$. If by theoretical fiat we choose to regard S and $S^{\prime }$ as “the same system”—e.g., we treat an organism as “the same thing” after it ingests a meal—we can conclude via *Theorem 1* that the VFE experienced by that system has increased. Its VFE may, of course, thereafter decrease if the new interaction $H_{S^{\prime }E^{\prime }}$ proves more predictable than the previous $H_{SE}$.

Theorem 1 has a significant consequence for the FEP: no system can reliably predict, from the information available on its MB, whether it has reached its NESS. Indeed, to do so would require reliably predicting that its environment has reached its respective NESS, a feat explicitly ruled out by the MB condition.

For the far-from equilibrium systems that predominate in biology, the active inference formalism transforms an intractable integration problem of a global thermodynamic potential into a tractable integration over a probabilistic model of how a system “thinks” it should behave, i.e., a model of the behavior of the states of its MB. To illustrate this, recall Equation (1) and consider how a system learns about, and hence enhances its ability to predict the behavior of, its environment. At each instant, the state m of the MB can be considered the “next” element of the training set T. The internal state i of S can be considered to implement some function f; indeed this f is, in the language of the FEP, S’s generative model at that instant. If S is an artificial neural network, for example, f is implemented by the node transfer function (which can be taken to be identical for all nodes) and the weight values. In the early stages of learning, i.e., far from the NESS, significant variation in m will induce significant variation in f and hence in i. An active system, e.g., an organism or an autonomous robot, will modify m by behaving, i.e., acting on E to provoke some response. As the predictive capability of f improves, i.e., as i approaches in NESS, changes in m will invoke smaller changes in f, with no further changes in f (i.e., no further learning) as the limit in which “perfect prediction” and the NESS are achieved. 

This principle of active inference has recently been applied directly to biological self-organization, beyond its initial applications in neuroscience, from lower levels of biological complexity such as a primordial soup [28], to higher levels of cells and tissues during morphogenesis [29,31]. Indeed, it has been shown to map quite well to the evolution of larger-scale, higher-complexity systems. If each of a collection of small-scale systems (e.g., cells or individual organisms) can minimize its own, individual VFE (and hence maximize its predictive success) by closely associating with other small-scale systems, a larger-scale system (e.g., a multi-cellular organism or an organismal community) will emerge; such a larger-scale system will remain stable as a macroscopic, collective entity, and hence have a well-defined MB at its larger scale, but only if its dynamics successfully minimize VFE at its larger scale [79]. In this context, natural selection at each scale can be viewed as Bayesian model selection (also referred to as structure learning, because the structure of the generative model itself is evolved and learned [80]) at that scale [30]. This view of natural selection as Bayesian belief updating can be traced back to Fisher’s fundamental theorem of natural selection [81], which equates the rate of increase in fitness to the genetic variance in fitness [81,82]. Fisher’s theorem itself has been cast in information theoretic terms by S.A. Frank in relating the rate of fitness increase to the amount of information that a population acquires about its environment based on the price equation [83], which itself can readily be shown to be an instance of Bayes’ theorem [81]. For an evolving organism, each instantaneous interaction with the environment serves as a “training example” for the generative model as discussed above [84]. How selection pressure generates new neuronal structures underlying new generative models as its own variational free energy minimizing process has been shown in [75]. The fundamental operations of expanding and adapting generative models in response to selection pressure have been outlined in [85] and form a gradual emergence of complexity, balancing model complexity and accuracy demands from increased environmental dynamic complexity driving evolution. 

The limit in which selection (i.e., the modification of the generative model) stops and the NESS is achieved is the limit of “perfect adaptation” to the local environment. This process can be formulated in the traditional language of fitness as follows. As “fitness” can be considered a probability of future existence—i.e., of survival and/or reproduction—it can be expressed as a (instantaneous) function $g:\;e,m,i\;\to \;\left[0,1\right]$, where, as above, e, m, and i, are (instantaneous) external, blanket, and internal states, respectively (hence $\left(m,i\right)$ is the total organism state). Perfect predictions minimize VFE and maximize survival probability; indeed, the fundamental prediction of any system is its own future existence [32]. Hence, with appropriate normalization, we can write $g\left(e,m,i\right)=1-\delta$, where δ is the total prediction error of the generative model implemented by the internal state i when exposed to the external state e as represented on the blanket state m (see also [86], where this is made time-dependent using a path-integral formalism). Systems with large values of δ are unlikely to survive/reproduce; those with small δ are more likely to survive/reproduce. It is important, moreover, to note that nothing in the theory guarantees the stability of emergent large-scale structures; indeed, many organisms are only facultatively multi-cellular, and as human history amply demonstrates, multi-organism communities often collapse. It is also important to note that while MBs are often associated intuitively with cell membranes or other biophysically implemented spatial boundaries, the MB itself is a decompositional boundary in an abstract state space that may or may not assign spatial degrees of freedom to every state.

It is not our intention here to develop a general theory of evolution as Bayesian satisficing; for some initial steps in this direction, see [84]. Our interest is in a particular aspect of this process, the development of metaprocessor architectures as a way of obtaining more predictive efficacy at a smaller free-energy cost. We also explicitly set aside the question of how one agent identifies another and tracks the other’s identity through time. Identification and tracking require that some criterial component of the identified system maintains an NESS; detailed models of the identification and tracking process are developed in [33,69].

## 3. Results

In the following sections, we will introduce a formal description of metacognition in an evolutionary setting and discuss how different time scales in the interaction dynamics of simple, well-known and -studied systems induce metacognition in these systems. In Section 3.1, we will formally show how metacognitive abilities confer a fitness advantage in an evolutionary setting based on information theoretic energy considerations. In Section 3.2, we start by reviewing some different computational examples of such simple interaction networks and discuss how complex system dynamics in each case necessitate some form of metacognition. We will first look at multi-agent active inference networks, which are simultaneously the most general models of interaction, and exhibit the most explicit form of metacognition we discuss here. Then, we move on to the more specific, established computational frameworks of multi-agent/network models, including Lotka–Volterra-based systems of predator–prey interactions, coupled genetic algorithms, and GANs, highlighting and exploring their various roles in modeling coadaptation and coevolution. While we will mostly focus on two agent/network interactions, we note that an arbitrary number n of interactions can, while often technically challenging, be decomposed, without the loss of generality, into n binary interactions, each between one of the n agents and the remaining n−1 agents considered as a single system. This follows directly from the Markov blanket condition; at a deeper level, it follows from the associativity of vector-space decompositions [73]. Finally, in Section 3.3, we will discuss how spatio-temporally coarse-grained structures emerge naturally in any resource-limited system with sufficiently complex interaction dynamics.

### 3.1. Formal Investigation of Metacognition in Evolution

Our primary claim in this paper is that selection on multiple timescales induces metacognition. We can make this claim precise by employing the general formalism for a finite system S interacting with a finite environment E via a Markov blanket MB developed in Section 2.3. *Interesting* interactions involve nontrivial information processing between reading the MB **ℬ** (i.e., sensation) and writing on **ℬ** (i.e., action) on the part of both S and E. Landauer’s principle [87,88] imposes a free-energy cost of at least $\varepsilon =k_{B}T\;ln2$ per erased bit (and hence per rewritten bit) on (classical) information processing, where $k_{B}$ is Boltzmann’s constant and T>0 is the temperature of the system performing the computing. Note that this is a lower limit; a realistic system will have a per-bit processing cost of $\varepsilon =\beta k_{B}T$, with β>ln2 a measure of thermodynamic efficiency. As this free energy must be sourced, and the waste heat dissipated, through **ℬ** asymptotically, we assume for simplicity that it is sourced and dissipated through **ℬ** on every cycle. We can, therefore, represent **ℬ** as in Figure 2b. The bits allocated to free-energy acquisition and waste heat dissipation are uninformative to S, as they are burned as fuel for computation. The bits allocated to action by S are similarly uninformative to S, although they are informative to E. The bits allocated to sensation by S are informative to S and serve as the inputs to S’s computations. This distinction between informative and uninformative bits is often not made explicitly (such as in Figure 3), where the flow of free energy as incoming fuel and outgoing waste is ignored; however, it is required whenever the joint system SE is thermodynamically closed [33]. To emphasize this distinction, we will use s for informative sensory states and a for states encoding “interesting” actions, i.e., actions other than waste heat dissipation.

With this formalism, we can place lower limits on the free energy cost of the computational processes that determine S’s actions on E, given various assumptions about E’s actions on S and S’s computational architecture. As noted above, E’s actions on S implement natural selection. The characteristic times of E’s actions are, therefore, the characteristic times of the selective pressures that S faces.

In the simplest case, E acts on S via some stochastic process that varies randomly with a characteristic timescale τ. As S has, in principle, no access to the internal dynamics of E except through **ℬ**, from S’s perspective, τ is the response time of E to actions by S. We assume for simplicity that S’s response time is also τ, i.e., that S is capable of computing its next action in time τ from detecting E’s current action.

We also assume for simplicity that S devotes the same number n of bits on **ℬ** to sensation and action, i.e., S computes a function $f:\;{\left\{0,1\right\}}^{n}\;\to \;{\left\{0,1\right\}}^{n}$, and that this function depends, in general, on the values of all n input bits (i.e., S does not ignore informative bits). If computing this function requires m steps, including all relevant classical memory write operations, then S has a free-energy cost of nmε per sensation–action cycle.

Now consider a situation in which E acts on S via two independent stochastic processes with two different characteristic timescales, $\tau_{1}$ and $\tau_{2}$, and assume that S detects and processes the inputs from these two environmental processes independently. Assume also that S’s thermodynamic efficiency is the same for detecting and processing the inputs from these two environmental processes. In this case we can factor the state s  into two sectors, $s_{1}$ and $s_{2}$, encoding $n_{1}$ and $n_{2}$ bits, respectively, the states of which vary at $\tau_{1}\;$ and $\tau_{2}$, respectively. Suppose now that $\tau_{1}=\tau$ and $\tau_{2}>\tau_{1}$, i.e., suppose S experiences independent selective pressures with two different characteristic timescales. In this case, S’s free-energy cost to process only the fast-varying inputs from $s_{1\;}$ is $n_{1}m\varepsilon$ per sensation–action cycle, while S’s free-energy cost to process only the slowly varying inputs from $s_{2\;}$ is $\left(\tau /\tau_{2}\right)\;n_{2}m\varepsilon$ per sensation–action cycle. Hence, S’s total free-energy cost for processing the two inputs separately is:(4)$$\Xi =\left(n_{1}+\left(\tau /\tau_{2}\right)n_{2}\right)m\varepsilon +\Delta ,$$
where Δ is the overhead cost, which must also be sourced as free energy from E, of detecting the change in $s_{2}$ and devoting resources to processing it. Whether Ξ<nmε, i.e., whether processing the inputs separately is more energy efficient than processing all n input bits together on every cycle, clearly depends on the value of Δ. Hence, we can state:

**Theorem** **2.***If informative inputs vary at different timescales*$\tau_{1}$ and $\tau_{2}$, $\tau_{2}>\tau_{1}$
*, processing the inputs separately is energetically advantageous provided its cost*
$\Delta <\left(1-\tau /\tau_{2}\right)n_{2}m\varepsilon$
*, where*
$n_{2}$
*is the number of bits of the slowly varying input,*
m
*is the number of computational steps for each input bit, and ε is the per-bit thermodynamic cost.*

**Proof.** The cost of processing the inputs together on each cycle is nmε; hence, $\left(1-\tau /\tau_{2}\right)n_{2}m\varepsilon$ is the energy saved by processing the inputs separately if the Δ=0. Provided Δ remains below this number, separate processing is advantageous. □

Separate processing of inputs is, clearly, a simple form of metaprocessing; the metaprocessor is the “switch” that detects a change in $s_{2}$ and devotes processing resources to it; hence, Δ is the overhead cost of metaprocessing. For example, the sector $s_{1}$ may encode inputs from an “object” that S identifies in E, while $s_{2}$ encodes the “background” surrounding the object, i.e., the entire rest of E. Distinguishing an object from its environmental background requires distinct rates of state change; typically, objects change their states faster than the background does, e.g., by moving [89]. We can, therefore, consider segmenting objects from their environmental backgrounds a simple and evolutionarily ancient metacognitive function.

Other things being equal, higher energetic efficiency confers a fitness advantage. Hence, we can state:

**Corollary** **1.***Organisms implementing metacognition will generically dominate organisms that do not*.

**Proof.** Other things being equal, the fitness advantage of any energy cost savings due to metaprocessing is exponentially amplified. Hence, in N generations, the more efficient organisms will have a fitness advantage of ${\left(1+\left(nm\varepsilon -\Xi \right)\right)}^{N}$. □

Selective pressures with multiple timescales, therefore, generically favor metacognition. The fact that even simple organisms such as *E. coli* employ metaprocessing architectures is, in this case, not surprising. Organisms favor metaprocessing architectures for the same reason software engineers do: because they are more efficient.

To highlight the biological relevance of optimizing bit-wise information processing energy costs, it is noteworthy that individual cells and single-cell organisms can, and must, in order to survive, detect and compute very small differences in information [90]. For example, in bacterial chemotaxis, it has been shown that fast-moving *E. coli* can respond to concentration changes as low as 3.2 nM [91] in as little as one second, the time available to evaluate ligand concentrations between “tumbles” [92]. As this concentration value corresponds to only about three ligand molecules in the volume of the cell, assumed to be one femtoliter, this actually suggest single-molecule detection [90]. As Aquino and Endres have shown, biochemical signaling precision against a background of large thermal noise and in far-from-equilibrium systems is enhanced by active cellular processes such as receptor internalization, allowing quick removal and subsequent internalization-rate-dependent time averaging of sensed ligands [90]. As such processes of endocytosis almost always require some form of energy consumption such as through ATP [93], we should expect single cells and single-cell organisms to evolve metaprocessing architectures based on the considerations above. Indeed, for this exact reason, Landauer’s principle has recently been applied to, and confirmed in, cellular biochemical networks for steady-state computations of ligand concentrations [94,95].

In the next section, we examine several well-studied architectures, showing in each case how they use metaprocessing to improve efficiency.

### 3.2. General Models of Two-System Interaction with Selection across Different Time Scales

#### 3.2.1. Multi-Agent Active Inference Networks

When coupling active-inference systems together to study multi-agent behavior, the first thing to notice is that actions from one agent are natively coupled into the sensory inputs of the others and vice versa without need for further extension, as opposed to the other modeling frameworks below. This is because of the way the MB partitions the systems into active, sensory, internal, and external states from the start: active states of one subsystem are directly coupled with the sensory states of others, while the external (also called hidden) states are made up of the internal states of others, as internal states between agents are hidden from each other by the intervening MB. What is left to state explicitly is exactly how the updating of beliefs and subsequent actions of one agent affect those of others in the same context of coadaptation and coevolution as above.

Variational free energy, the term to be minimized at the center of active inference, places an upper bound on (logarithmic) model evidence (i.e., the likelihood of sensory data with respect to an internal model), and thereby on the dispersion of a particle’s internal states and their Markov blanket. Because in a multi-agent system the reduction of the dispersion of sensory, and therefore internal states, necessitates the minimization of the dispersion of active states of other agents, we can equate this goal as each agent working to limit the behavioral flexibility of the others. When different agents have the same type of generative model, the system will quickly, after the random initialization of prior beliefs, converge on a stable, quasi-static configuration. This was the case in our previous work on morphogenesis as Bayesian inference, where the generative model encoded a target morphology as the goal structure [29,31]. In the coevolutionary systems we are discussing in this paper, however, this is no longer the case. In such a system with different generative models, the task becomes for each agent to learn the generative model of its environment. Fortunately, we do not need to reinvent the mechanism of learning in a purely active inference framework, but can instead integrate deep neuronal networks into an active inference model, which is what has recently been achieved under the deep active inference framework [86]. This framework deals specifically with the problem of an agent having to contend with environmental dynamics to which it not only does not have direct access to, but also whose underlying functional form does not have to coincide with the functional form of the agent’s generative model of the world. To solve this problem, deep active inference employs deep and recurrent neural networks [96,97] as flexible functional approximators for the internal dynamics of the agent, allowing the agent’s generative model to learn a good approximation of true world dynamics [86]. On top of that, this framework employs evolution strategies, an extension to genetic algorithms that focus on unbiasedness and adaptive control of parameters [98]. These evolution strategies estimate gradients on the variational free energy bound on the average agent’s surprise over a population of agents [86], surprise here meaning how unexpected sensory inputs vary compared to an agent’s prior beliefs, which is what is being minimized in active inference.

A concrete application of deep active inference to adaptive behavior in a human setting has recently been implemented in the form of studying emotional valence [99]. This paper shows that the type of deep hierarchical structure employed in deep active inference can allow an agent to hold and utilize information in working memory as a basis for future valenced actions. This is relevant to this discussion for two reasons:

(1) We have recently shown that the concept of valence goes far beyond neuronal systems, is at the core of the fundamental drive to self-preservation underlying organisms as living agents, and is very compatible with the basic computational approaches to learning [100,101].

(2) By maintaining internal valence representations in a deep active inference model, the ensuing affective agent is capable of optimizing confidence in action selection preemptively. These valence representations can then in turn be optimized by leveraging the (Bayes-optimal) updating term for subjective fitness, tracking changes in fitness estimates and lending a sign to otherwise unsigned divergences between predictions and outcomes [99]. This formulation of affective inference then corresponds to implicit metacognition [99] and is exactly the necessary capability that will emerge in our considerations of coevolution below.

#### 3.2.2. Predator–Prey Models

Complex temporal ecological interaction has been identified as a driving force behind evolution. Despite the extensive evolutionary records supporting this notion, the actual prediction of coevolutionary outcomes is extremely difficult because of a lack of access to the experimental records of the actual dynamic interactions of different species in concert with their environment. One of the simplest ways to computationally model such a coevolutionary interaction is by building on predator–prey models, which predict the population dynamics of two different species in a predator–prey relationship over time. Predator–prey models are typically based on Lotka–Volterra equations of the form:(5)$$\frac{dx}{dt}=\alpha x-\beta xy\;\frac{dy}{dt}=\delta xy-\gamma y,$$
initially proposed in the context of chemical reactions [102], where x and y are the number of prey and predator organisms, and $\frac{dx}{dt}$ and $\frac{dy}{dt}$ the instantaneous growth rates of the two populations, respectively, with *α*, *β*, *γ*, *δ* as positive real parameters describing the individual traits determining the interaction between the two species.

This system of linear partial differential equations results, in nontrivial parameter settings, in oscillating dynamics where the growth peaks of the predator population cause a collapse of the prey population, which in turn results in a collapse of the predator population, followed by a rebound of the prey population and so on (cf. Figure 4). This simplified cyclic dependence, however, is built on the assumption of species life history traits being fixed over ecologically relevant time scales [103]. Moreover, once this population-level framework is extended to (co-)evolution, implicit inbuilt timescales of the system such as life cycle and environmental shifts quickly necessitate the explicit formulation and analysis of these time scales for the system to be solvable [104,105].

A first step towards introducing the ability of either population to adapt as part of an evolutionary process was provided by the Rosenzweig–MacArthur extension [107], which includes density-dependent population growth and a functional response. Because this extension allows for self-limitation in prey and handling limitation in the predator populations, prey adaptation over evolutionary time scales can then be included in stabilizing the system in more complex cases such as more than one coexisting predator species [108].

Coevolution can then be incorporated following [103] by allowing the predator and prey traits determined through the parameters α and β (and with them implicitly δ and γ) from Equation (5) to vary over time:(6)$$\frac{d\alpha }{dt}=V_{a}\left(\alpha \right)\frac{\partial }{\partial \alpha_{i}}\left[\frac{1}{x}\;\frac{dx}{dt}\right]{│}_{\alpha_{i}=\alpha \;}\;\frac{d\beta }{dt}=V_{b}\left(\beta \right)\frac{\partial }{\partial \beta_{i}}\left[\frac{1}{y}\;\frac{dy}{dt}\right]{│}_{\beta_{i}=\beta }\;.$$

Here, $V_{a}\left(\alpha \right)$ and $V_{b}\left(\beta \right)$ denote the genetic trait variances of predator and prey, while the remaining terms in these trait equations determine the individual fitness gradients for the traits, whose derivatives are taken with respect to an individual’s phenotype ($\alpha_{i}$ and $\beta_{i}$).

Much work has been carried out on finding stabilities, as well as critical transitions in the population dynamics that are the basis of evolutionary shifts, using, among others, bifurcation analysis to show that different time scales in prey and predator populations reduce the model dimensionality enough to allow for different behavioral strategies to influence system stability and persistence necessary for evolution to act upon [104,105,109,110].

The temporal patterns emerging from such treatments of the predator–prey model crucially help explain experimental evidence in the ecology of temporal (as well as spatial) separation in terms of behaviors [111,112] and life cycles [113,114], in which different species have seemingly learned the temporal patterns of their cohabitants. Indeed, when analyzing predator–prey models in the context of different ecological and evolutionary timescales, alternating, periodic patterns of different prey and predator pairs have been found, and it indeed has been shown that such evolutionary timescales are essential for the maintenance of different species in a system [115].

It is in this experimental context that computational models of reinforcement and deep reinforcement learning have been applied to the within-lifetime behavior, as well as the evolution of new behavior and physical properties on longer time scales [106,116,117,118,119]. Specifically, it has been shown that the reinforcement learning of predators is beneficial to the stability of the ecosystem, while the learning of prey makes the ecosystem oscillate and leads to a higher risk of extinction for predators [118]. Crucially, the same research shows that the co-reinforcement learning of predators and prey leads to predators finding a more suitable way to survive with their prey, meaning the number of predators is more stable and greater than when only predators or only prey learn [118], further supporting the notion that each species perceives and adapts to the temporal patterns of other species.

In other words, by highlighting the clear adaptive and learning components in coevolving interactions and modeling approaches thereof, we identify components of a metaprocessing architecture that provide us with a higher-level, predictive view of coevolution. To further elaborate on the idea of coevolving species, we next review work on coupled genetic algorithms, which allows us to better study temporal adaptation over long time periods.

#### 3.2.3. Coupled Genetic Algorithms

While Darwin is credited with originating the core concept of evolution through natural selection, it was not until the 20th century that scientists discovered the various molecular processes that created the genetic diversity upon which natural selection could occur [120]. Once the fundamental processes of mutation and crossover were sufficiently illuminated, evolution could be reduced to a surprisingly simple yet efficient algorithm, which Holland first formulated as a tool not just to simulate evolution, but also to solve other computational problems by operating as a general-purpose classifier system [121]. The basic concept of a standard evolutionary algorithm is to represent a population of individual organisms as part of a species to be evolved (or solution to be found) as sequences, or strings, of individual units called bits (which can represent genes or even individual nucleotides when modeling actual evolution) that are in initially randomized states of a predefined set (cf. Figure 5). Furthermore, one must define a scorable attribute to the string called the fitness function that can be computed at each time step. In the case of evolution, this fitness function describes how fit, i.e., how probable to appear in the next generation, any given phenotype is given the organism’s environment. Then, at each simulation step, a certain amount of mutation (random changing of states of a limited number of bits), as well as crossover (switching of states between different regions or different strings) is performed, and the fitness function for the different outcomes is calculated. Outcomes that decrease fitness are eliminated, while outcomes that increase it will be passed to the next iteration, corresponding to the next generation in an evolutionary setting. Thereby, outcomes that increase fitness appear more frequent in subsequent iterations, allowing the solution space to be sampled very quickly yet broadly [122,123].

The two biggest limitations of genetic algorithms are that (1) they can readily arrest in local minima of the solution space and (2) they are very sensitive in their efficiency to parameters governing the mutation and crossover probabilities, as well as the definition of the fitness function itself, leading to a lot of problem-specific finetuning of the algorithm [122,124,125]. An important step towards remedying this problem and creating more efficient and robust genetic algorithms has been taken in the form of introducing adaptation to the genetic algorithm, in which either the probabilities of mutation and crossover are made dependent on the fitness values of previous generations [126], or the fitness function itself is dynamically varied according to environmental constraints [127]. This straightforward computational finding has wide-ranging implications for biological evolution, specifically explaining the source of the rapid adaptation of species in novel environments, such as is observed in invasive species [126].

More importantly for this work, it is this inclusion of an adaptive property to a genetic algorithm that allows us to study coevolution and the emergence of the temporal patterns in which we are interested. The most fundamental aspect of coevolution is that the fitness functions of each species will invariably depend on the evolution of other organisms, creating the need to include time-varying and tracking functions into the genetic algorithm as above, otherwise the convergence cannot be achieved on realistic time scales [128,129].

We have already implicitly shown how a need for coevolution emerges in the interaction between different species through the example of predator–prey modeling in the previous section, and now we turn to how this is achieved using coupled genetic algorithms. Coupled genetic algorithms are referred to as such because they couple the fitness function of different coevolving individuals and systems [130,131,132,133], thereby each computing the selection function for the other, exactly as shown in predator–prey interactions [134,135]. We note that if behavior during the lifetime of individuals significantly affects fitness and therefore evolution, this coupling can replace the “all at once” sequential fitness calculation of a standard genetic algorithm with a more partial, but continuous, fitness evaluation, which occurs during the entire lifetime of an individual, called lifetime fitness evaluation [136].

Interestingly, it has been shown that there are two ways to maintain stability and successful long-term adaptation in a coevolutionary algorithm of a predator–prey type system [134]: First, one can include a sufficiently long memory in the subspecies of how fit and therefore well-adapted it has been compared to the other species in previous lifetimes. Second, one can balance the coevolution by making the reproduction rate dependent on the performance of a population, leading to different reproduction rates in both populations, just as we have seen in the previous section [113,114]. Both solutions implicitly cause each subspecies to learn the temporal patterns of the coevolving species.

Furthermore, research on coupled fitness landscapes in coevolving systems suggests that evolutionary dynamics through natural selection would favor individuals that achieve higher coevolutionary adaptive success, if the sustained average fitness of coevolving partners depends upon the ruggedness and coupling of their landscapes [137]. The emergence of characteristic time scales in this framework, consistent with our theoretical and experimental observations above, is then described as a necessary condition (acting via selective pressure) towards achieving evolutionary adaptability. This is achieved by forcing a system to a “liquid” phase state “poised at the edge of chaos”, where cascades (or avalanches) of evolutionary change propagate across all spatial scales in a power law distribution of characteristic cascade size and frequency [137].

In order to make this implicit learning of other coevolving strategies explicit, we now turn to a third type of model, called coupled GANs, which will allow us to directly express and study agents that learn each other’s evolutionary strategies.

#### 3.2.4. Coupled Generative Adversarial Networks

A basic generative adversarial network consists of a generator and a discriminator.

The objective of the generative model is to synthesize data, while the objective of the discriminative model is to distinguish real data from fake data. The key novelty in this approach as compared to many other machine learning approaches, both supervised and unsupervised, is that the learning of the generator is indirect through the discriminator, as it is not actually trying to maximize how close its generated outputs are to real outputs, but instead aims to counteract the ability of the discriminator to tell the difference between fake and real inputs. Crucially, because the discriminator itself is also updated dynamically, this becomes an adaptive, competitive process that necessitates a lot fewer restrictions on the type of generative model we begin with. However, it still allows the generator to focus its generation on a smaller manifold close to the real data, which is why they can generate more precise outputs than other models [138]. One way to apply a generative adversarial network in a dynamic adaptive setting similar to the simulations of coadapting and coevolving species above has recently been provided by Talas et al., where the generative network corresponded to prey developing better camouflage, while the predator was represented by the discriminating network trying to discern the prey from its environment [139]. As would be expected in a real, yet constrained to one physical dimension, predator–prey evolutionary arms race, the simulation ends in either an equilibrium or the “extinction” of one or both components, corresponding to the discriminator finding 100% of the generated targets on every iteration.

In order to apply this method to more complex, i.e., multi-agent (more species modeled by more generative and discriminative networks) and/or multi-directional (one species being modeled by both a generative and a discriminating network, making adaptation truly bidirectional), one only needs to couple multiple GANs together (cf. Figure 6). Indeed, coupled GANs are at the forefront of novel research on finding computational solutions to problems of learning joint distributions, where tuples (pairs) of corresponding data sets are not provided, resulting in a more biologically realistic case of unsupervised domain learning [140,141]. Indeed, in such frameworks, the learning of spatio-temporal separation and necessary memory conditions that we have seen earlier can be modeled explicitly by employing Spatio-Temporally Coupled Generative Adversarial Networks (STC-GANs). These networks learn powerful temporal representations by extracting rich dynamic features and high-level spatial contexts via a spatio-temporal encoder in a future frame generation model [142]. Here, STC-GANs are used for predictive scene parsing, which captures contextual appearance information and dynamic temporal representation from prior frames to generate future scene parsing results. In other words, this transforms the implicit learning of temporal patterns in coevolutionary contexts into an explicit problem of predicting future adaptive patterns from coevolving species in an environment.

These examples all illustrate the core result of Section 3.1: that metacognition emerges when selection acts on multiple timescales. We now consider the emergence of coarse-grained structures in metacognition, showing when and how they emerge in natural systems.

### 3.3. Spatio-Temporally Coarse-Grained Structures Emerge Naturally in Any Resource-Limited System with Sufficiently Complex Interaction Dynamics

We have seen multiple computational reasons to include an explicit representation of time and spatial scales in interacting model systems based on tractability arguments and have seen how such representations emerge in model systems with built-in time scales. We now consider how these representations emerge and function in general.

We start by pointing out that the originally biologically motivated and now heavily utilized computational algorithm we discussed above (Section 3.2.2) is already implicitly scaled (or chunked): cross-over in genetic algorithms. Besides the obvious biological motivation of genes as chunks, it is a lot more computationally efficient to enhance mutation-only searches with cross-over, as it allows the algorithm to search deep in parts of the space defined by fixed subsequences. The main advantage of the inbuilt coarse-graininess which the cross-over provides is that it allows the search algorithm to maintain search diversity by constantly combining novel mutations across the size of the genome, reducing the risk of landing on local minima. In other words, by coarse-graining the different search operators across different time scales, we drastically increase the efficiency of the algorithm in limited computational times by interleaving depth-first (i.e., mutation) and breadth-first (i.e., cross-over) searches. From this perspective, it makes sense that the biological genome would evolve to incorporate multiple scales of organizations, because otherwise adaptive evolution within time constraints relevant to selection pressure would be unlikely.

Another example of scaled spatio-temporal structures is presented by morphogenesis in multi-cellular organisms. Morphogenesis, by definition, concerns the establishment of the shape of an entire organism, meaning that successful morphogenetic processes are, when viewed as a search path in a morphogenetic free-energy landscape, not primarily about individual cell behavior. What this means is that at different stages of a morphogenetic process, individual cells migrate, rearrange, or differentiate not necessarily based on individual cellular decisions, but based on environmental feedback integrated from large numbers of surrounding cells. Therefore, morphogenesis is about morphogenetic fields, signaling concentrations that are chunked into corresponding cell types [143,144]. In this view, the spatial scale introduced by multi-cellularity is an optimal solution to the problem of obtaining a large, functionally complex shape. Moreover, this scaled structure is also represented in the temporal scales of morphogenesis, where fast time scales continually integrate into emerging slower time scales as multi-cellularity increases with morphogenesis.

Finally, we turn back to the question of evolution by highlighting some key properties of multi-level selection theory. We have already seen in our discussions about co-evolution that learning and anticipation are practically required in order for coadapting and coevolving systems to stabilize in sufficiently complex temporal interactions. It seems therefore unlikely that higher levels of multi-evolved systems would emerge in a pure depth-first search without any coarse-graining of spatio-temporal scales. In fact, recent work in formally linking evolutionary processes to principles of learning seems very promising [145] in explaining how modularity evolves in dynamic environments [146], and consequentially how those structures emerge from learned past environments by generalizing to new environments [147]. Indeed, evolutionary and developmental processes can be viewed as executing the same dynamics, which is well-described by the active inference framework, just at different spatial and temporal scales [84,148].

Lastly, while highly intuitive, we highlight the fact that despite the above considerations on the emergence of time- and space-scaled structures in a closed, controlled environment, the actual background of all living things is of course highly structured in time and space by purely physical constraints. This includes the seasonal [149], lunar [150], and day–night cycles [151,152], as well as various weather and geological patterns that build a scaled foundation upon which evolution is built.

## 4. Conclusions

The central aim of this work was to highlight how the emergence of multi-scale processing is a necessary requirement for many evolutionary scenarios and constitutes a form of metacognition based on an architectural definition of metaprocessing.

We showed that metacognitive systems must evolve naturally when fitness landscapes vary on multiple time scales. We then illustrated this by reviewing coevolutionary modeling frameworks based on predator–prey models and genetic algorithms. Here, quasi-stable coevolution was only possible when implicit or explicit considerations on time scales and methods of learning were included.

We described how new approaches in generative adversarial networks have the potential to link the adaptive learning components of a coevolving system more natively and tightly to a predictive framework in the case of spatio-temporal coupled generative adversarial networks, while keeping a very intuitive form of competitive selection.

We showed that the new emerging application of active inference to biological systems of various complexity comes with an inbuilt formalized structure of active and sensory states that form an interface (the Markov blanket) between the internal (belief-encoding and hence predictive) and external (environmental) states. This interface is, from a design perspective, an application programming interface connecting two virtual machines, with the coding capacity of the application programming interface determining the extent to which the two machines/agents can communicate. Thus, from remarkably simple assumptions, we produce a direct formulation of metacognition in terms of learning the generative model of other agents in a multi-agent active inference framework. Furthermore, incorporating deep learning directly into active inference in recent work [86,99] has the potential to supply a complete model of adaptive, multi-scaled coevolution that would allow one to link complex learning outcomes to remarkably simple, valence-based goal structures that fit well into more established homeostatic considerations on biological drive [100].

Finally, we discussed how time-scaled and coarse-grained structures must emerge naturally in any resource-limited system with sufficiently complex interaction dynamics. This is because, as we proved, under generic assumptions, metacognition is more energetically efficient, and hence confers higher fitness, than purely object-level cognition when selection operates at multiple timescales. We underlined this point by showing that existing computational approaches to coevolution all incorporate, implicitly or explicitly, forms of metacognition without which coadaptation, and hence (co-)evolution, cannot be efficiently achieved. This realization is furthermore strengthened by the observation that the physical environment by which evolution is constrained is already highly structured in time and space.

We thereby conclude that, given these physical constraints on time and resources, evolution could only have converged on solutions that contain metacognitive aspects. This is because the architectural concept of metaprocessing underlying our use of the term metacognition is essential for evolution to have resulted in such diverse and highly structured ecosystems as we observe today.

Crucially, from this follows that we should expect to find various distinct time scales in the structure and behavior of evolved organisms, giving us an important insight into their dynamic interactions with other species, as well providing us with a vital tool with which to manipulate coadaptive outcomes.

The exciting challenge of the coming years will be to elucidate exactly how specific temporal scales manifest in various levels of a coevolving biological system, so that one day we can predict evolutionary outcomes with enough accuracy to act, when necessary, before the collapse of an ecosystem, including our own.

## Figures and Tables

**Figure 1 entropy-24-00601-f001:**
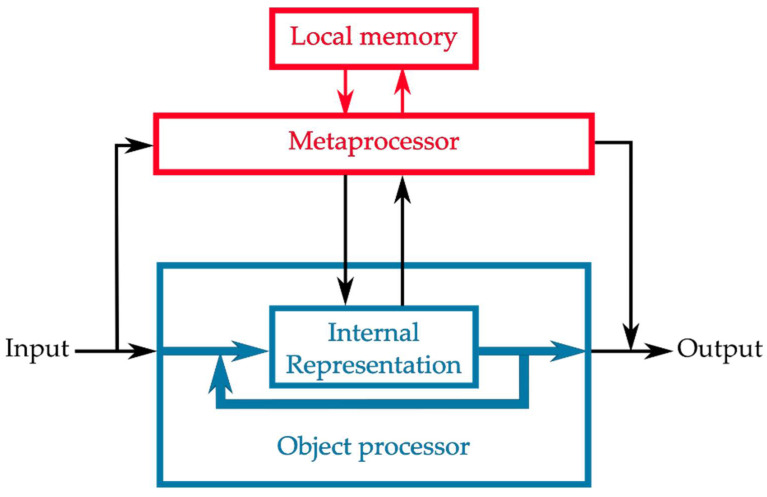
Generic architecture of a metaprocessor. A metaprocessor (red) regulates an object processor (blue) both internally and externally. The metaprocessor requires its own local memory of the object processor’s behavior.

**Figure 2 entropy-24-00601-f002:**
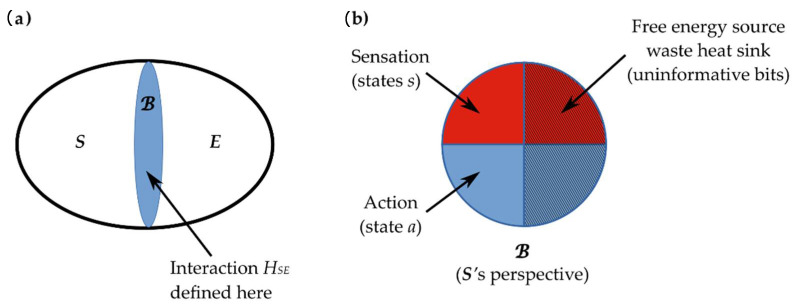
General form of bipartite interaction dynamics in the absence of quantum entanglement. (**a**) Finite systems ***S*** and ***E*** interact via a boundary **ℬ** that serves as an MB. (**b**) From ***S***’s perspective, **ℬ** comprises sensory bits (in state *s*, red shading) that are inputs to ***S***’s computations and action bits (in state *a*, blue shading) that are ***S***’s outputs. When the free-energy costs of computation are taken into account, as in Section 3.1 below, some input bits must be allocated to fuel for computation and some output bits must be allocated to waste heat dissipation (hatched area); these bits are uninformative to ***S***.

**Figure 3 entropy-24-00601-f003:**
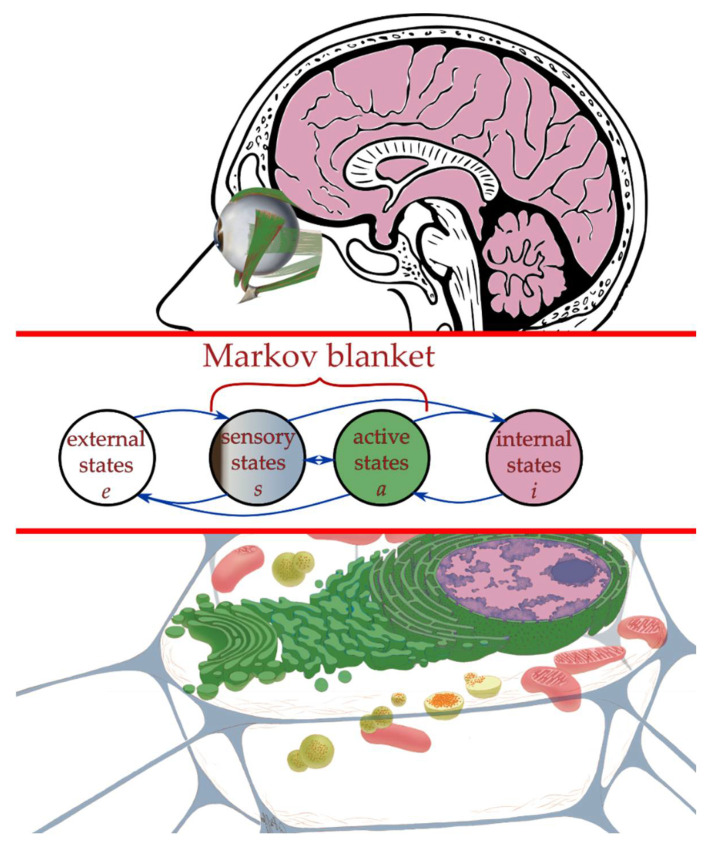
Markov blanket state separation in active inference. **Middle**: The internal (pink) and external states (white) of each agent are separated by a Markov blanket, comprised of sensory (grey) and active (green) states. This architecture can be applied to various forms of information processing, two of which are shown above and below. By associating the gradient flows of the Markov blanket partition with Bayesian belief updating, self-organization of internal states—in response to sensory fluctuations—can be thought of as perception, while active states couple internal states back to hidden external states vicariously, to provide a mathematical formulation of action and behavior. **Top**: Visual processing. Internal states are made up of the brain, which directs movement of the eye through abductor muscles as active states. Sensory states as the photoreceptors in the eye perceive the external states in the visual field of view. **Bottom**: Transcriptional machinery. Internal states are given by the gene expression levels plus epigenetic modifications. Intracellular components such as ribosomes, smooth and rough endoplasmic reticulum, and Golgi apparatus implement protein translation and delivery as the active states. Sensory states correspond to the surface states of the cell membrane, such as ligand receptors, ion channel states, and gap junctions between cells. External states are associated with extracellular concentrations and the states of other cells. Image credits: The cell schematic has been adapted from an image by Judith Stoffer supplied at the National Institute of General Medical Sciences, NIH (CC BY-NC 2.0). The brain schematic has been adapted from an image of a head cross-section on openclipart.org by Kevin David Pointon under a public domain license, and an image of an eye with abductor muscles from Wikimedia by Patrick J. Lynch (CC BY-NC 2.5).

**Figure 4 entropy-24-00601-f004:**
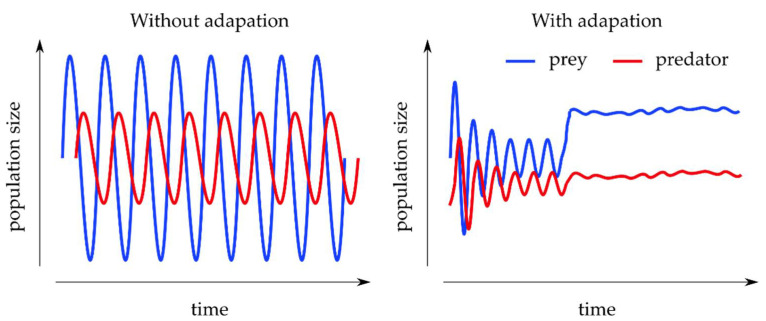
Schematic of adaptive learning in predator–prey models. **Left**: A typical predator–prey model simulation based on Equation (5), characterized by persistent population size oscillations of both predator and prey, where peaks in prey precede peaks in predator populations. **Right**: Predator–prey model simulations with learned adaptive coevolution of both predator and prey based on Equation (6) and the work done by Park et al. [106], characterized by initially dampened oscillations followed by only marginally oscillating, stable population sizes of both predator and prey.

**Figure 5 entropy-24-00601-f005:**
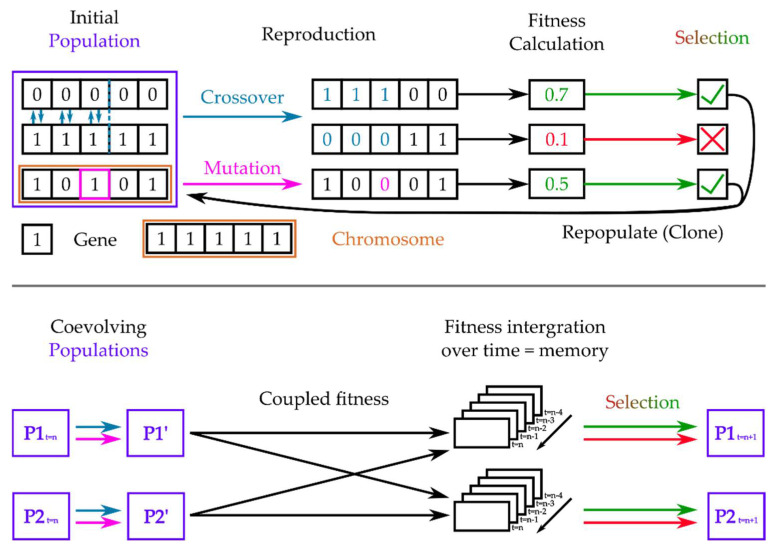
Coupled Genetic Algorithm Schematic. **Top**: Schematic of a standard genetic algorithm. An initial population (purple) of individuals or chromosomes (orange), consisting of individual bits or genes (black), reproduce with crossover of random sets of genes and mutations. Fitness according to a fixed fitness function is calculated for each chromosome, after which individuals with highest fitness are selected and cloned to form a new population. After multiple iterations of this process, the algorithm stops when no higher fitness can be generated in any new individual. **Bottom**: Schematic of a coupled genetic algorithm. Coevolving populations undergo reproduction as usual but are then coupled in the computation of each other’s fitness function, which in turn are variable and can incorporate integration of past fitness functions as a form of memory, at which point learning methods can be implemented.

**Figure 6 entropy-24-00601-f006:**
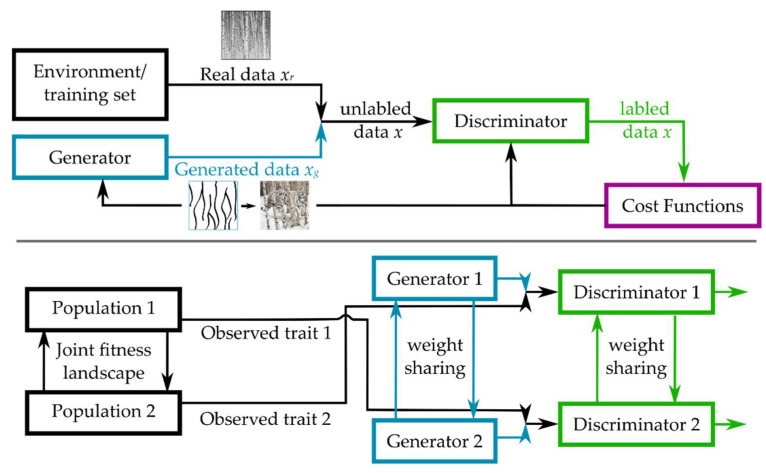
Schematic of Coupled Generative Adversarial Networks. **Top**: Schematic of a standard generative adversarial network applied to the problem of camouflage. A generator (blue) neural network generates data *x**_g_* with the goal of emulating the potential distributions of real data *x**_r_* as best as possible. In this example, inspired by Talas et al. [139], the generator generates stripe patterns that, when emerged in the habitat of a Siberian tiger, are aimed at fooling the discriminator in not being able to discern generated tiger patterns from tree patterns in the background. A second neural network called discriminator receives unlabeled data from both real and generated distributions and aims to label them correctly by origin. **Bottom**: Schematic of a coupled generative adversarial network. When fitness landscapes of two populations are coupled, each generative adversarial network type agent tries to internally generate and discern trait patterns that correspond to the observed traits of the other population, thereby learning and predicting the traits of the other population. Because in a coupled generative adversarial network weights between the first few generator layers, and last few discriminator layers are shared, this configuration allows the coupled generative adversarial network to learn a joint fitness distribution underlying traits without correspondence supervision [140]. Tiger and birch tree images are adapted from images taken from flickr.com (last accessed 02/09/2022) under the CC BY-NC-SA 2.0 license.

## Data Availability

Not applicable.

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
