# Peer review of "Metacognition as a Consequence of Competing Evolutionary Time Scales"

_entropy, 2022, doi:10.3390/e24050601_

Round 1

Reviewer 1 Report

Reviews for entropy-1612202

I think the manuscript is suitable for Entropy. I would request the authors to include a new section of discussing the psychological, cognitive, and neural basis of metacognition more deeply and broadly. They should also discuss/include from nonhuman animals such as nonhuman primates and rodents. Here are a list of researchers in these fields that the authors should include and reference.

Humans: Steve Fleming, Xiaohong Wan, Hakwan Lau

Nonhuman primates and neural basis: Kentaro Miyamoto

Rodents: Armin Lak, Adam Kepec

Given the ‘evolutionary’ aspect is mentioned (even in the manuscript title), a discussion on the nonhuman primate vs. humans literature should be warranted. Here are some work on these aspects, which the authors should reference.

    Hampton, R. R. Multiple demonstrations of metacognition in nonhumans: converging evidence or multiple mechanisms? Comp. Cogn. Behav. Rev. 4, 17-28 (2009).
    Miyamoto, K. et al. Causal neural network of metamemory for retrospection in primates. Science 355, 188-193 (2017).
    Middlebrooks, P. G. & Sommer, M. A. Metacognition in monkeys during an oculomotor task. J. Exp. Psychol. Learn. Mem. Cogn. 37, 325-337 (2011).
    Miyamoto, K., Setsuie, R., Osada, T. & Miyashita, Y. Reversible silencing of the frontopolar cortex selectively impairs metacognitive judgment on non-experience in primates. Neuron 97, 980-989.e6 (2018).

On humans:

    Fleming, S. M., Weil, R. S., Nagy, Z., Dolan, R. J. & Rees, G. Relating introspective accuracy to individual differences in brain structure. Science 329, 1541-1543 (2010).
    Fleming, S. M., Huijgen, J. & Dolan, R. J. Prefrontal contributions to metacognition in perceptual decision making. J. Neurosci. 32, 6117-6125 (2012).
    McCurdy, L. Y. et al. Anatomical coupling between distinct metacognitive systems for memory and visual perception. J. Neurosci. 33, 1897-1906 (2013).
    Fleming, S. M., Ryu, J., Golfinos, J. G. & Blackmon, K. E. Domain-specific impairment in metacognitive accuracy following anterior prefrontal lesions. Brain 137, 2811-2822 (2014).
    Ye, Q. et al. Individual susceptibility to TMS affirms the precuneal role in meta-memory upon recollection. Brain Struct Funct 224, 2407–2419 (2019).

Author Response

Thank you for these references.  We have included a new Sect. 2.1 discussing metacognition in higher organisms – including nonmammals – more broadly.  We explicitly note that the phylogenetic distribution of metacognitive abilities remains controversial, and hence the evolution of metacognition remains substantially unclear.  This lack of clarity motivates our functional-architectural approach.  We also relate metacognition more closely to theories of consciousness as requested by Reviewer 2.

Reviewer 2 Report

This study aims to highlight how the emergence of multi-scale processing underlying adaptation is a requirement for many evolutionary scenarios and why evolution could be seen as a form of metacognition. Based on the machine learning definition of meta-processing, the researchers presented a generic architecture in which a meta-processor regulates an object processor both internally and externally utilizing its own local memory of the object processor’s behavior. By looking at coevolutionary modeling frameworks based on predator-prey models and genetic algorithms, they showed that metacognition emerges whenever fitness functions vary on multiple timescales. This study concludes that metacognition is far from human-specific. On the contrary, it can be expected at every level of the biological organization to address the same problem, that of dealing effectively with uncertainties having different temporospatial scales.

I think this paper is interesting. It addresses a great topic highlighting aspects that are commonly overlooked in previous studies. Among the strengths of this article, I can highlight the strong research questions, deep knowledge in this particular field, originality, novelty and high quality. Researchers do not hesitate to utilize complex scientific theories from different research fields aiming at a full-scale approach. Thoughts are well-organized and flow smoothly. The ideas are linked clearly and cohesively. The abstract gives the reader a good idea of what to expect from the paper. Language and presentation are clear and adequate. The headings enable the reader to understand the main points of the paper and to follow its structure. In general, this study is in line with the journal’s guidelines.

I would like to suggest some minor revisions that I believe the paper needs to be publishable in the journal. Although most terms, theories and concepts mentioned in the study are adequately defined, I believe that the theory of metacognition should be further analyzed especially in the Introduction. I suggest the 8X8 layered model of metacognition which is in line with various aspects of this paper  (Drigas & Mitsea, 2020; 2021).

In addition, the authors might consider making a brief reference to the concepts of consciousness and motivation (see Satti et al., 2021; Paoletti and Ben-Soussan, 2020; Maslow, 1943).

 I conclude that this study has the significant potential to be an article with added value in the international literature. According to my point of view, the above suggestions would positively contribute to the acceptance of this article.

Thus, I recommend that this paper be accepted after minor revision.

Drigas, A., & Mitsea, E. (2021). 8 Pillars X 8 Layers Model of Metacognition: Educational Strategies, Exercises &Trainings. International Journal of Online & Biomedical Engineering17(8).

Drigas, A., & Mitsea, E. (2020). The 8 Pillars of Metacognition. International Journal of Emerging Technologies in Learning (iJET), 15(21), 162-178

Sattin, D., Magnani, F. G., Bartesaghi, L., Caputo, M., Fittipaldo, A. V., Cacciatore, M., & Leonardi, M. (2021). Theoretical Models of Consciousness: A Scoping Review. Brain sciences, 11(5), 535.

Paoletti, P., & Ben-Soussan, T. D. (2020). Reflections on inner and outer silence and consciousness without contents according to the sphere model of consciousness. Frontiers in Psychology, 1807.

Maslow, A. H. (1943). A theory of human motivation. Psychological Review, 50, 370–396.

Author Response

Thank you for these references.  We now mention the detailed model of Drigas & Mitsea (2020) in the Introduction, in order to contrast it with the broader idea of metacognition employed here.  We then consider models of consciousness explicitly in the new Sect. 2.1.  While some models of consciousness (e.g. GNW and HOT) are naturally interpreted in terms of metacognition, others (e.g. IIT) are not.  This lack of agreement motivates our focus on a purely-functional definition of metacognition and architectural definition of metaprocessing.

Reviewer 3 Report

In this study, the authors applied active inference to biological systems provided with an inbuilt structure of active and sensory states that form an interface or the Markov blanket. They attempted to formulate evolution as metacognition in terms of learning the generative model of other agents under a multi-agent active inference framework. They concluded that evolution could only have converged on solutions that contain metacognitive aspects because metaprocessing are essential for evolution of diverse and highly structured ecosystems.

I agree with an importance of characterising evolution as a multi-scale optimisation process and appreciate the authors’ attempt to model evolution using active inference. However, I don’t think the presented results support their conclusion. Despite the mathematical claims on properties of evolutionary process, this paper involves no mathematical proof at all. Some assumptions are unjustified.

Major concerns:

  1. The Markov blanket cannot be applied to the evolutionary process in the manner that the authors discuss because it is a non-stationary process. From the perspective of surprise minimisation, coevolution might work to limit the internal states of others, as the authors expected. However, this coevolution process is in a non-steady state and thus the conditional independency between internal and external states given blanket states doesn’t hold. Please note that the Markov blanket and equation (1) are valid only under a (non-equilibrium) steady state. They are applicable when the evolution is saturated and reaches a steady state, but not when the evolution is progressing. This means that the mathematics introduced in Sec. 2.1 is not suitable to explain the main results in Sec. 3. The authors should distinguish between steady state and non-steady state clearly, and use active inference under appropriate assumptions.

  1. Although the authors state that multi-agent active inference is the most general model (L216), indeed the main results in Secs. 3.2–3.4 have no direct relation to active inference. For Fig. 3, Refs. [52] and [54] are not papers on active inference. To explain the result under active inference, the authors have to commit to specific generative model and variational free energy that correspond to the system defined by equation (3) and show that equation (4) is derived as the gradient descent on variational free energy. Claiming as if active inference explains the result without showing rigorous derivation constitutes an abuse of mathematical terminologies. Similarly, how the results in Figs. 4 and 5 relates to active inference is not explained.

  1. Related to the above comment, the authors need to create a separate Methods section and describe the mathematical definitions of the example setups shown in Results. They should clearly define a specific generative model for each example and show that the algorithms used in the simulations are derived as the gradient descent on variational free energy. This is necessary to make this paper scientifically sound.

  1. Although the authors state that “the mathematical frameworks representing natural selection […] have the same logical structure as Bayesian belief updating [26]” (L198–200), Bayesian belief updating is a continuous updating process, whereas natural selection is implemented by discrete mutation and crossover. They are qualitatively different. I believe that Ref. [26] doesn’t state that Bayesian belief updating and natural selection have the same logical structure. Neither the previous nor this work proves that natural selection minimises free energy.

  1. The novelty of the presented simulation results is unclear. For example, the simulated behaviour of the Lotka-Volterra model and stabilisation of the oscillation were already observed in Ref. [52], and equation (4) was proposed in Ref. [54]. What is the significance of stabilising the oscillation using equation (4)?

Other comments:

L43: The definition of metacognition in this paper is not very clear. I agree on the importance of hierarchical control, but don’t understand why it should be called metacognition. There are counterexamples: a hierarchical control can exist even in a simple machine that has no cognitive function.

L46: According to conventional control theory, the thermostat example simply means that the thermostat is the controller and the furnace is the plant, where the furnace should be treated as the external state, not as an object processor. It is unclear why this can be an example of ‘meta’cognition.

L49: Provide the definition of fitness functions explicitly.

L181–L184: I believe m in equation (1) is unnecessary. Please note that the Markov blanket is not a variable, so it should not be an argument of the conditional probability. The phrase “a generative model implicit in the Markov blanket m” makes no sense and should be deleted.

L229: sytem --> system

Fig. 3 legend: equation (4) should be equation (3), and equation (5) should be equation (4).

Author Response

In this study, the authors applied active inference to biological systems provided with an inbuilt structure of active and sensory states that form an interface or the Markov blanket. They attempted to formulate evolution as metacognition in terms of learning the generative model of other agents under a multi-agent active inference framework. They concluded that evolution could only have converged on solutions that contain metacognitive aspects because metaprocessing are essential for evolution of diverse and highly structured ecosystems.

     We have revised both the Introduction and the text that follows to emphasize that our objective is to show that evolution generically yields metaprocessor architectures and metacognition as a function when selective pressures vary on multiple timescales.  We do not, in particular, claim to show that (biological, natural-selection driven) evolution itself is an active inference process, though we have argued elsewhere that biological evolution can be described in terms of active inference at multiple identifiable scales.  As the reviewer points out in #1 below, this requires defining the “system,” the “environment,” and the interaction between them at each scale in a way that removes non-stationarity at that scale, i.e. renders the Markov blanket well-defined.  While this is possible with suitable formal manipulation whenever the joint system is finite, this analysis is outside present scope and we do not discuss it here.

I agree with an importance of characterising evolution as a multi-scale optimisation process and appreciate the authors’ attempt to model evolution using active inference. However, I don’t think the presented results support their conclusion. Despite the mathematical claims on properties of evolutionary process, this paper involves no mathematical proof at all. Some assumptions are unjustified.

          We have clarified our assumptions throughout and now provide a constructive proof of our main claim in Sect. 3.1.

Major concerns:

  1. The Markov blanket cannot be applied to the evolutionary process in the manner that the authors discuss because it is a non-stationary process. From the perspective of surprise minimisation, coevolution might work to limit the internal states of others, as the authors expected. However, this coevolution process is in a non-steady state and thus the conditional independency between internal and external states given blanket states doesn’t hold. Please note that the Markov blanket and equation (1) are valid only under a (non-equilibrium) steady state. They are applicable when the evolution is saturated and reaches a steady state, but not when the evolution is progressing. This means that the mathematics introduced in Sec. 2.1 is not suitable to explain the main results in Sec. 3. The authors should distinguish between steady state and non-steady state clearly, and use active inference under

appropriate assumptions.

     The evolutionary process along a phylogenetic path from some organism as root to a different organism as leaf may be non-stationary as stated here.  Our goal, as emphasized above, is not to characterize such paths in terms of active inference, but rather to show that they generically yield metaprocessor architectures and metacognition as a function at the leaf nodes.       We have clarified the conditions under which a Markov blanket is well-defined, and hence Eq. 2 and 3 hold, in Section 2.3.  In Section 3.2.1, we explicitly consider agents with well-defined Markov blankets.  Each agent interacts with its entire surroundings, including all the other agents.  This has the immediate consequence that specific, i.e. side-effect free, interactions between agents are in general not well-defined.  Indeed in such settings, it is not required that the agents can identify each other such.  Biological cells, for example, do not have to identify each other as cells to interact (as shown by their ability to interact also with electrodes, inorganic scaffolds, etc.).

  1. Although the authors state that multi-agent active inference is the most general model (L216), indeed the main results in Secs. 3.2–3.4 have no direct relation to active inference. For Fig. 3, Refs. [52] and [54] are not papers on active inference. To explain the result under active inference, the authors have to commit to specific generative model and variational free energy that correspond to the system defined by equation (3) and show that equation (4) is derived as the gradient descent on variational free energy. Claiming as if active inference explains the result without showing rigorous derivation constitutes an abuse of mathematical terminologies. Similarly, how the results in Figs. 4 and 5 relates to active inference is not explained.

      We have now clarified the text to indicate that we do not claim that Sections 3.2.2 – 3.2.4 are examples of active inference, though as noted above, they could be reformulated in this way.  They are rather examples that illustrate our main point, that exposure to selective pressures on multiple timescales generically yields metaprocessor architectures and metacognition as a function.

  1. Related to the above comment, the authors need to create a separate Methods section and describe the mathematical definitions of the example setups shown in Results. They should clearly define a specific generative model for each example and show that the algorithms used in the simulations are derived as the gradient descent on variational free energy.  This is necessary to make this paper scientifically sound.

     The simulations in Sections 3.2.2 – 3.2.4 are those of the cited authors, as reported in the cited papers.  The methods employed are described in the respective papers.  As noted above, we do not claim to reformulate these in terms of active inference or generative models.  What we add to the discussion is that these examples all exhibit metacognition as a function.  They serve, therefore, as supporting empirical evidence for our conclusion.

  1. Although the authors state that “the mathematical frameworks representing natural selection [...] have the same logical structure as Bayesian belief updating [26]” (L198–200), Bayesian belief updating is a continuous updating process, whereas natural selection is implemented by discrete mutation and crossover. They are qualitatively different. I believe that Ref. [26] doesn’t state that Bayesian belief updating and natural selection have the same logical structure. Neither the previous nor this work proves that natural selection minimises free energy.

     The relevant text at the end of Sect. 2.3 has been revised. 

  1. The novelty of the presented simulation results is unclear. For example, the simulated behaviour of the Lotka-Volterra model and stabilisation of the oscillation were already observed in Ref. [52], and equation (4) was proposed in Ref. [54]. What is the significance of stabilising the oscillation using equation (4)?

        As noted above, in Sect. 3.2.2 we are reviewing the results presented in the cited references, and pointing out that they can be interpreted as exhibiting metacognition as defined here. We have made this overall more explicit in the text now, while also highlighting the novel capability we gain from our new metacognitive interpretation of existing work on adaptation and learning in predator prey models.

Other comments:

L43: The definition of metacognition in this paper is not very clear. I agree on the importance of hierarchical control, but don’t understand why it should be called metacognition. There are counterexamples: a hierarchical control can exist even in a simple machine that has no cognitive function.

        We have revised the Introduction to clarify our definitions of “metacognition” and “metaprocessor.”  The former is purely functional, and depends on no philosophical assumptions about what systems might or might not be “cognitive”; we now explicitly set any such questions aside, referencing prior work on this topic.

L46: According to conventional control theory, the thermostat example simply means that the thermostat is the controller and the furnace is the plant, where the furnace should be treated as the external state, not as an object processor. It is unclear why this can be an example of ‘meta’cognition.

          Our definition of a metaprocessor is purely architectural; hence a thermostat is a metaprocessor for a furnace as object processor if the “system of interest” boundary is drawn so as to include both thermostat and furnace.  If thermostat and furnace are considered to be separate systems, the thermostat is simply a switch, and the furnace is a system regulated by an external switch.  These alternative ways of drawing the “system of interest” boundary make no difference to the state of the world; they are interpretative, not ontic.  This is now made explicit in the Introduction.

L49: Provide the definition of fitness functions explicitly.

        We have provided an explicit, generic definition of the fitness function in Sec. 2.3.  The specific fitness functions used in the simulation studies reviewed in Sect. 3.2.2 – 3.2.4 are, as noted above, those of the cited authors.

L181–L184: I believe m in equation (1) is unnecessary. Please note that the Markov blanket is not a variable, so it should not be an argument of the conditional probability. The phrase “a generative model implicit in the Markov blanket m” makes no sense and should be deleted.

          Both the previously-redundant notation and the relevant phrase have been corrected.  In the notation in Sect. 2.3, m = (s, a).  We expand the idea of m in Sect. 3.1 to include purely thermodynamic exchange; here (s,a) is the informative component.

L229: sytem --> system

        Thanks.  Corrected.

Fig. 3 legend: equation (4) should be equation (3), and equation (5) should be equation (4).

         Thank you for this correction, you are right. In the revised manuscript, we ended up with an additional equation 3 in the new section 3.1 so that the references in Fig. 3. ended up being (4) and (5) again.

Round 2

Reviewer 1 Report

 After reviewing the revised manuscript, I would like to comment further by suggesting the authors include a more balanced coverage of the NHP literature. Below is a snapshot of the vast literature, which I suggest the authors try to include by all means.  Otherwise, I do not have further comments and would recommend 'acceptance' when these are done. 

Robert Hampton, Kentaro Miyamoto, Marc A. Sommer, Kwok 

    Hampton, R. R. Multiple demonstrations of metacognition in nonhumans: converging evidence or multiple mechanisms? Comp. Cogn. Behav. Rev. 4, 17-28 (2009). 
    Miyamoto, K. et al. Causal neural network of metamemory for retrospection in primates. Science 355, 188-193 (2017). 
    Middlebrooks, P. G. & Sommer, M. A. Metacognition in monkeys during an oculomotor task. J. Exp. Psychol. Learn. Mem. Cogn. 37, 325-337 (2011). 
    Miyamoto, K., Setsuie, R., Osada, T. & Miyashita, Y. Reversible silencing of the frontopolar cortex selectively impairs metacognitive judgment on non-experience in primates. Neuron 97, 980-989.e6 (2018). 
    Kwok, S. C., Cai, Y. & Buckley, M. J. Mnemonic introspection in macaques is dependent on superior dorsolateral prefrontal cortex but not orbitofrontal cortex. J. Neurosci. 39, 5922-5934 (2019). 
    Cai Y, Jin Z, Zhai C, Wang H, Wang J, Tang Y, Kwok SC Time-sensitive prefrontal involvement in associating confidence with task performance illustrates metacognitive introspection in monkeys. DOI: https://www.biorxiv.org/content/10.1101/2021.11.30.470665v2.abstract 

Author Response

Reviewer 1:

After reviewing the revised manuscript, I would like to comment further by suggesting the authors include a more balanced coverage of the NHP literature. Below is a snapshot of the vast literature, which I suggest the authors try to include by all means.  Otherwise, I do not have further comments and would recommend 'acceptance' when these are done.

        Done. Thank you for pointing out these further references, we have included them in Sect. 2.1.

Reviewer 3 Report

I would like to thank the authors for writing clearly about the conditions under which the Markov blanket can be applied. However, I think that the revision rather makes it clearer that introducing the Markov blanket does not help to support the main claim. Although they state in the response that "In Section 3.2.1, we explicitly consider agents with well-defined Markov blankets", when the learning is progressing, the system is in a non-steady state. Thus, the fact that "the task becomes for each agent to learn the generative model of its environment" (L495) rather suggests that the Markov blanket is not well defined under such a problem setup.

Further, given the authors' answer that "The simulations in Sections 3.2.2 – 3.2.4 are those of the cited authors, as reported in the cited papers" and this study only proposes a new interpretation, this paper seems not to be an original research paper but rather a perspective paper.

Suddenly the revised section 3.1 describes quantum information, which I think has little relationship to the other parts of this paper. The mathematical rigour of the section 3.1 is weak, which undermines the credibility of this paper.

I don't think that introducing quantum information help to support the authors' argument because the interpretation such as "E’s actions on S implement natural selection" (L365) violates the steady state assumption anyway. If it is a stationary process, the offspring distribution should be equal to the parent distribution to keep a steady state, which means that no selection can occur.

The later part of section 3.1 simply says that if parts of sensory inputs change slowly, using a slow processor is sufficient to process them, which is more energy efficient than using a rapid processor as long as the overhead cost is small enough. However, I don't think that this directly relates to the Landauer’s principle. According to the Landauer’s principle, forgetting or losing 1 bit information emits heat of at least kT ln2. Considering that the information per unit time is n_2/\tau_2 for s_2, the processor can forget at most n_2 bit information for every \tau_2 time, which is determined by the external milieu condition and thus unchanged despite that the authors change the time constant of the processor from \tau_1 to \tau_2. This indicates that the definition of energy cost in equation 3 is not consistent with the Landauer’s principle. Does this equation satisfy the first principle of thermodynamics when considering the energy balance?

L348: In relation to biology, the body temperature of organisms is high and large thermal fluctuations exist. There are also large fluctuations in the activity of sensory cells and synapses. Under such noisy conditions, it is impossible for organisms to compute the small difference in information considered in the Landauer’s principle. This is also the case for machines in real world.

L389: The free-energy cost in equation 3 seems to have no direct relation to variational free energy defined in section 2.3. By construction, they are qualitatively different from each other.

Round 3

Reviewer 3 Report

I would like to thank the authors for the effort in revising the manuscript. As for the novelty of the results, I would like to leave the judgement to the editors and thus will not comment further. I only point out major concerns about scientific quality:

As for the steady state assumption, I agree with that "Maintaining a NESS is criterial for system identifiability by an external observer; however, only the component that is criterial for that observer needs to maintain a NESS, allowing other components to vary." However, this doesn't answer to my concern. My previous comment intended that, when one focuses on the learning time scale, it needs to maintain a NESS, meaning that while allowing other components (e.g., inference time scale) to vary, one cannot use variational free energy in equation (1) to model the learning process because it is a non-steady state when the learning is progressing. Similarly, when focusing on the evolution time scale, modeling the process of natural selection is outside the scope of equation (1). Further, it seems obvious that S and E are entangled when S infers E by construction; thus, I believe that "the assumption that S and E are not entangled" is contradictory.

As for equation (3), I still think that the Landauer's principle doesn't derive equation (3). Because the authors consider the case where the equality holds, I believe that they consider a quasi-static process. Under the quasi-static process, W_meas + W_eras = 2kTln2 = W_ext holds for 1 bit, where W_meas is work required for measurement, W_eras is work required for erasure, and W_ext is work extracted by the cycle (Sagawa, Ueda, Phys Rev Lett, 2008, 2009). The extracted work W_ext = 2kTln2 is used for measurement and erasure W_meas + W_eras. Because S doesn't change its entropy or free energy between the beginning and end of each cycle, I believe that the authors' statement such that "S’s total free-energy cost for processing the two inputs separately is" given by equation (3) is incorrect or at least inaccurate. Equation (3) is not the free energy change in S. Also, equation (3) seems not to be the extracted work or work for measurement and erasure. It is also unclear how and from where work \Delta is extracted. If it is not a quasi-static process, the Landauer's principle does not guarantee the equality.
